# Serological detection of *Mycobacterium Tuberculosis* complex infection in multiple hosts by One Universal ELISA

Liang Sun[1,2,3,4,5], Yingyu Chen[2,3,4,5], Ping Yi[2,3,4,5], Li Yang[2,3,4,5], Yu Yan[2,3,4,5], Kailun Zhang[2,3,4,5], Qiaoying Zeng📵[1]\*, Aizhen Guo📵[2,3,4,5]\*

**1** Laboratory of Veterinary Microbiology, College of Veterinary Medicine, Gansu Agricultural University, Lanzhou, China, **2** The State Key Laboratory of Agricultural Microbiology, Huazhong Agricultural University, Wuhan, China, **3** College of Veterinary Medicine, Huazhong Agricultural University, Wuhan, China, **4** National Animal Tuberculosis Para-Reference Laboratory (Wuhan), Ministry of Agriculture and Rural Affairs of the People's Republic of China, Huazhong Agricultural University, Wuhan, China, **5** Hubei International Scientific and Technological Cooperation Base of Veterinary Epidemiology, International Research Center for Animal Disease, Ministry of Science and Technology of the People's Republic of China, Wuhan, China

\* zengqy@gsau.edu.cn (QZ); aizhen@mail.hzau.edu.cn (AG)

**Data Availability Statement:** All relevant data are within the manuscript and its Supporting Information files.

## Abstract

Tuberculosis (TB), a contagious disease mainly caused by *Mycobacterium tuberculosis* (*M. tb*), *Mycobacterium bovis* (*M. bovis*), and *Mycobacterium caprae* (*M. caprae*), poses a major global threat to the health of humans and many species of animals. Developing an ante-mortem detection technique for different species would be of significance in improving the surveillance employing a One Health strategy. To achieve this goal, a universal indirect ELISA was established for serologically detecting *Mycobacterium tuberculosis* complex infection for multiple live hosts by using a fusion protein of MPB70, MPB83, ESAT6, and CFP10 common in *M. tb*, *M. bovis*, and *M. caprae* as the coating antigen (MMEC) and HRP-labeled fusion protein A and G as a secondary antibody. After testing the known positive and negative sera, the receiver operating characteristic curves were constructed to decide the cut-off values. Then, the diagnostic sensitivity and specificity of MMEC/AG-iELISA were determined as 100.00% (95% CI: 96.90%, 100.00%) and 100.00% (95% CI: 98.44%, 100.00%) for *M. bovis* infection of cattle, 100.00% (95% CI: 95.00%, 100.00%) and 100.0% (95% CI: 96.80%, 100.00%) for *M. bovis* infection of sheep, 90.74% (95% CI: 80.09%, 95.98%) and 98.63% (95% CI: 95.14%, 99.76%) for *M. bovis* infection of cervids, 100.00% (95% CI: 15.81%, 100.00%) and 98.81% (95% CI: 93.54%, 99.97%) for *M. bovis* infection of monkeys, 100.00% (95% CI: 86.82%, 100.00%) and 94.85% (95% CI: 91.22%, 97.03%) for *M. tb* infection of humans. Furthermore, this MMEC/AG-iELISA likely detects *M. caprae* infection in roe deer. Thus this method has a promising application in serological TB surveillance for multiple animal species thereby providing evidence for taking further action in TB control.

**Funding:** This work is funded by China Agriculture Research System of MOF and MARA (Beef/yaks) (#CARS-37).

**Competing interests:** The authors have declared that no competing interests exist.

# 1 Introduction

Tuberculosis (TB) is known as an important zoonotic infectious disease caused by various *Mycobacterium tuberculosis* complex (MTBC) members. The main members of MTBC include *Mycobacterium tuberculosis* (*M. tb*), *Mycobacterium bovis* (*M. bovis*), and *Mycobacterium caprae* (*M. caprae*), which share a wide range of hosts but maintain different host preferences [1]. For example, human TB (hTB) is mainly caused by *M. tb*, but *M. bovis* of animal origin can infect humans and cause hTB either. According to the Global Tuberculosis Report in 2020 by the World Health Organization, there were about 10.0 million new TB cases and 1.4 million new deaths in 2019 [2]. Among the hTB cases, the proportions of cases caused by *M. bovis* infection varied in different countries and areas. For example, in Ethiopia, Tanzania, Turkey, and the west bank of Palestine, 17.0%, 26.1%, 5.3%, and 6.5% of hTB cases were reported to be caused by *M. bovis* respectively [3–8]. In addition, *M. caprae* infection accounted for 0.3% of hTB cases in Spain [9]. On the other hand, more and more publications reported *M. tb* infection in animals especially bovines [10–12]. Sheep TB cases are mainly caused by *M. caprae* and *M. bovis* transmitted from cattle under the condition of cohabitation or sharing grazing land. In Spain, sheep cohabiting with infected cattle or goats were reported to have a prevalence of 50.44% caused by the infection with *M. bovis* or *M. caprae* [13]. Additionally, wild animals such as deer and monkeys are also susceptible to these three MTBC members [14,15]. Therefore, OIE added the infection of bovids, cervids, goats, and New World camelids by *M. bovis*, *M. tb*, and *M. caprae* into the disease list of the Terrestrial Animal Health Code since 2019. In May 2014, the World Health Assembly passed a resolution of the post-2015 End TB Strategy with its ambitious targets to be fully realized in 2035. However, the maintenance of MTBC infection in the multi-host system is a great barrier to achieving this goal worldwide. To quickly investigate the epidemiological status of TB in many different species of live animals, it is ideal to have one common universal test for all.

Currently, commercial detection methods including reagents and kits are mainly developed for bovine TB (bTB) and hTB. Diagnosis of TB for other animals caused by *M. bovis*, *M. caprae*, and *M. tb* in sheep, goats, cervids, monkeys, and some other wild animals close to humans and domestic animals is usually based on the conventional methods such as the tuberculin skin or eye tests, mycobacterial acid-fast staining and culture, postmortem examination for typical tubercles and histopathological analysis, but these conventional detection methods either have low-sensitivities or low-specificities, or are unsuitable for the examination of live animals [16]. Therefore, a universal method to detect MTBC infection for multiple live hosts infected by three common MTBC members is urgently needed. Antibody detection is an ideal choice to identify the risk of TB transmission in multiple species due to its advantages of simple, easy, rapid, and high-throughput screening.

However, antibody detection based on a single antigen is usually low sensitivity in TB diagnosis. It could be overcome by the combination of different antigens produced at different growth stages of MTBC [17,18]. The potential candidate antigens include MPB70, MPB83, ESAT6, CFP10, P22, MPB64, Arc1, MPB59, MPBCF, and others [17,19], and the antibody responses against these antigens have been identified respectively in different hosts such as bovines, deer, humans, monkeys, wild boars, cervids (sika deer, red deer, roe deer, fallow deer), Spanish ibices, mouflons, and elephants [14,18–23].

Another limitation of the antibody test with a traditional ELISA lies in the species specificity of the secondary antibodies which varies with host species. One solution addressing this is to develop an antigen sandwiched ELISA such as the commercial MPB83 sandwiched ELISA kit (INGEZIM Tuberculosis DR, Ingenasa, Spain) which has been validated in the detection of *M. bovis* infection in some wildlife (deer, badgers, wild boars, etc) and goats. However, as

mentioned above, by targeting a single antigen, it might not display sufficient sensitivity. Alternatively, an ELISA using a universal ligand of Ig such as *Staphylococcal* protein A and/or *Streptococcal* protein G and fused multiple antigens would overcome the limitation in species and sensitivity [24,25].

Therefore, this study aimed to develop a single universal serological detection method for MTBC infection named MMEC/AG-iELISA by using a fusion protein made from four specific MTBC proteins MPB70, MPB83, ESAT6, and CFP10 (MMEC) as the coating antigen and HRP-labeled fusion protein of *Staphylococcal* protein A and/or *Streptococcal* protein G (AG-HRP) as the secondary antibody to bind Ig of various hosts species. This MMEC/AG-iELISA was optimized to detect antibodies to MTBC infection in bovines, sheep, cervids, monkeys, and humans.

## 2 Materials and methods

### 2.1 Ethics statement

Animal experiments to collect sera were conducted in strict accordance with the Guide for the Care and Use of Laboratory Animals, Hubei Province, China. The protocols were approved by the Ethics Committee of Huazhong Agricultural University (agreement no. HZAUSH-2019-005).

### 2.2 Serum sample collection

The information of serum samples from various hosts is presented in Table 1. Human sera were collected previously by the Tuberculosis department, Wuhan Medical Treatment Center, and stored at -80˚C by this laboratory [26]. Briefly, the 24 human serum samples were collected from 24 TB patients who were firstly suggested by clinical signs and chest X-ray test, and then confirmed by *M. tb* isolation from sputum with mycobacterial culture and molecular typing. The hospital used Xpert to rapidly type *M. tb* and test drug resistance gene *rpoB* for the patient treatment, while we used multiplex PCR to further differentiate MTBC members for our research described as follows. In addition, the 233 human negative sera were collected from healthy volunteers who were negative in intradermal tuberculin test (induration area < 5 mm), chest X-ray test, and *in vitro* blood IFN-γ release test.

Bovine serum samples included the following: 120 positive sera were confirmed by the comparative intradermal tuberculin test, *in vitro* blood IFN-γ release test, antibody test to bTB kit (IDEXX, America), and ELISA kit to detect the antibody against *M. bovis* (KEQIAN, China); 242 negative sera were confirmed by the comparative intradermal tuberculin test, *in vitro* blood IFN-γ release test, and with the above mentioned ELISA kits; the samples of antisera against *M. paratuberculosis*, Bovine herpes virus 1, *Brucella abortus*, and bovine viral diarrhea virus (BVDV) were purchased from China Institute of Veterinary Drug Control; 17 positive sera were collected from *M. bovis*-infected cattle, the tissue with pathological change of the 17 cattle were collected in an abattoir. The collected samples were homogenized and decontaminated with 4% sodium hydroxide. After neutralization and centrifugation, the sediment is inoculated on 7H11 medium (Becton Dickinson, Franklin Lakes, NJ) for a minimum of 8 weeks at 37˚C with $CO_2$ for MTBC culture. When growth is visible, the genome of the clone was extracted by boiling. Then, typing of the MTBC members was conducted by multiplex PCR targeting seven genes *16S Rrna*, Rv0577, IS*1561'*, Rv1510, Rv1970, Rv3877/8, and Rv3120 as described before [27]. Briefly, each PCR mixture was prepared with 25 μL of PCR Master (Tsingke Biotechnology Co., Ltd., China), 20 μL of water, 2.5 μL of dimethyl sulfoxide, 1 μL of each primer at 20 μM, and 0.5 μL of DNA. PCR amplifications were performed with an initial denaturation step of 5 min at 94˚C followed by 35 cycles of 60 s at 94˚C, 60 s at 60˚C,

**Table 1. Information on positive and negative serum samples collected previously and used in this study.**

| Host species | MTBC members | Positive sera | | Negative sera | | References |
|---|---|---|---|---|---|---|
| | | No | Criteria for positive | No | Criteria for negative | |
| **Human** | *M. tb* | 24 | Suggested by clinical sign and chest X-ray test, and confirmed by *M. tb* isolation from sputum in a TB professional hospital | 233 | Clinical health and negative in intradermal tuberculin test (induration area < 5 mm), chest X-ray test, and *in vitro* blood IFN-γ release test | [26] |
| **Cattle** | *M. bovis* | 137 | 17 samples from cattle confirmed by bacterial culture and PCR typing with unknown antibody titers, 120 positive sera defined by comparative intradermal tuberculin test, *in vitro* blood IFN-γ release test, and antibody tests with commercial kits from companies IDEXX[1] and KEQIAN[2] | 242 | Negative in comparative intradermal tuberculin test, *in vitro* blood IFN-γ release test, and antibody tests with commercial kits from companies IDEXX and KEQIAN | [27,28] |
| **Sheep** | *M. bovis* | 46 | Antiserum from the sheep immunized with commercial PPD-B and determined as positive with PPD-ELISA | 113 | Negative in comparative intradermal test and PPD-ELISA | [24] |
| **Sika deer** | *M. bovis* | 52 | Positive in PPD-ELISA and the INGEZIM kit [3] | 146 | Negative in PPD-ELISA and the INGEZIM kit | [24] |
| **Macaque monkeys** | *M. bovis* | 2 | Positive in PPD-ELISA and upper eyelid intradermal PPD-B test | 84 | Negative in PPD-ELISA, upper eyelid intradermal PPD-B test, and the INGEZIM kit | [24] |
| **Pere David's deer** | *M. bovis* | 1 | *M. bovis* isolation from lung tubercles | -[4] | - | [27] |
| **Roe deer** | *M. caprae* | 1 | *M. caprae* isolation from an abscess in the right submaxillary face | - | - | [27] |
| | | 3 | Positive in PPD-ELISA and these sera were collected from other three roe deer cohabiting with the *M. caprae* infected roe deer | | | [24] |

[1]: *M. bovis* antibody test kit (IDEXX, America).

[2]: ELISA kit to detect the antibody against *M. bovis* (KEQIAN, China).

[3]: MPB83 antigen sandwich ELISA produced by INGEZIM Tuberculosis DR (Ingenasa, Spain).

[4]: No sample.

and 90 s at 72˚C, and ending with a final elongation step for 10 min at 72˚C. PCR products were visualized by 1% agarose gel electrophoresis and sequenced. Meanwhile, the genome of MTBC in collected tissue was also directly extracted with nucleic acid extraction instruments (TIANLONG, China), and the subspecies of the MTBC were typed by multiplex PCR [27]. By mycobacterial culture and multiplex PCR typing, all 17 cattle were determined to be infected with *M. bovis*. However, for the tissue direct PCR typing, animals 9, 13, and 14 showed negative results (S1 Table).

Then 113 negative sheep sera were collected from healthy sheep (Chinese Hu sheep) which were determined with the cervical intradermal comparative test with bovine and avian purified protein derivative (PPD) [28]. After clipping hair, cleaning the skin, and measuring skin-fold thickness, 2500 IU/0.1 mL PPD of *M. bovis* abbreviated as PPD-B (CZ VeterinariaS.A., Porriño, Spain) or 2500 IU/0.1 mL PPD of *M. avium* abbreviated as PPD-A (CZ VeterinariaS.A., Porriño, Spain) was injected on each side of the neck at an identical position in the center of the middle third of the neck with a 1 mL sterile syringe. Seventy-two hours later, the skin-fold thickness was measured again. The reaction was considered to be negative if the increase in skin-fold thickness at the PPD-B injection site was less than or equal to the increase in the skin-fold reaction at the PPD-A injection site, and without clinical signs, such as diffuse or extensive oedema, exudation, necrosis, pain or inflammation of the lymphatic ducts in that region or the lymph nodes [28]. Then, 46 positive sheep sera to PPD-B were prepared as follows: the TB negative sheep were subcutaneously inoculated with 2500 IU/0.1 mL PPD-B and boosted with the same dose at 30 days later in the center of the middle third of the neck with a

1 mL sterile syringe, the positive sera were collected one week later, and titers were determined by PPD-ELISA as previously described [24]. Besides, one TB negative sheep was immunized with the mixture of 10 mL recombinant fusion protein MMEC (0.2 mg/mL) and 10 mL Freund's complete adjuvant (Sigma, America), and boosted with the mixture of the same amount of MMEC and Freund's incomplete adjuvant (Sigma, America) 15 days later, the positive sera were collected one week later and used as a positive control. Meanwhile, the serum sample collected from the sheep before immunization with MMEC was used as a negative control.

Deer sera included: 52 positive sika deer sera and 146 negative sika deer sera determined by PPD-ELISA and the INGEZIM kit [24], one positive serum sample to *M. bovis* was collected from a Pere David's deer which was determined by clinical signs, pathological changes, myco-bacterial culture, and PCR typing [27] with the samples from the tuberculous nodule in the lung; one positive serum to *M. caprae* was collected from a roe deer which was determined by clinical symptoms, mycobacterial culture, and PCR typing [27] from the animal with the abscesses in the right submaxillary face, meanwhile three positive serum samples collected from other three roe deer in the same group were determined by PPD-ELISA.

Monkey sera included: 2 positive sera determined by upper eyelid intradermal PPD-B test and PPD-ELISA, 84 negative sera collected from the animals which were determined by upper eyelid intradermal PPD-B test, PPD-ELISA, and the INGEZIM kit [24].

## 2.3 Expression and purification of MMEC

The recombinant plasmid encoding fusion protein MMEC was previously constructed in our lab [29]. Briefly, the genes were amplified from the genomic DNA of *M. bovis* and linked by a linker (GSGGGGSGGGGSGS). Further, the fusion DNA fragment was inserted into pET28a (+) plasmid, which was transformed into *E. coli* BL21 (DE3) (TIANGEN, China) for expression [29]. MMEC was purified by Ni SepharoseTM 6 Fast Flow resin gravity column (GE health-care, America). The purity of MMEC was evaluated by 10% sodium dodecyl sulfate-polyacryl-amide gel electrophoresis (SDS-PAGE), and the concentration of MMEC was determined by Enhanced BCA Protein Assay Kit (Beyotime, China).

## 2.4 HRP labeled protein AG

A volume of 100 μL of 20 mg/mL HRP (Sigma, America) was mixed with 100 μL of 21 mg/mL NaIO4 (Sigma, America) and the mixture was incubated for 30 min at 4˚C, then 2 μL ethylene glycol (Sinopharm, China) was added, and the obtained mixture was incubated for 30 min at room temperature in the dark. The oxidized HRP was mixed with 1 mL of 2 mg/mL protein AG (Prime gene, China), and the mixture was incubated for 2 h at room temperature, then 20 μL of 20 mg/mL NaBH4 (Sinopharm, China) was added and incubated for 2 h at 4˚C. After-ward, the AG-HRP was dialyzed overnight in PBS and stored at -80˚C after being mixed with 50% glycerol.

The concentration of AG-HRP was determined by Enhanced BCA Protein Assay Kit. The titer of AG-HRP was determined as follows: 100 μL of MMEC (100 ng/mL) was added to the 96-well plate and incubated overnight at 4˚C, then 200 μL blocking buffer (1% fish gelatin (Sigma, America) in PBS) was added and incubated for 1 h at 37˚C. Whereafter, the plate was incubated with 100 μL of positive or negative control serum sample pre-diluted with blocking buffer at 1:10000 for 30 min at 37˚C. Then the plate was incubated with 100 μL of two-fold serially diluted AG-HRP (diluted with blocking buffer) for 30 min at 37˚C. The plate was washed six times with PBST (0.5% Tween-20 in PBS) after each step during this procedure. Afterward, the plate was incubated with 100 μL of SureBlue TMB microwell substrate (KPL,

America) for 10 min in the dark, then mixed with 100 μL of 1 N HCl to terminate the reaction. Subsequently, the optical density (OD) values at the wave length of 450 nm were measured with a spectrophotometer.

## 2.5 Establishment of MMEC/AG-iELISA

After the MMEC coating concentrations of 200 ng/mL, 100 ng/mL, 50 ng/mL, 25 ng/mL, and 12.5 ng/mL and AG-HRP concentrations of 4 μg/mL, 0.8 μg/mL, 0.4 μg/mL, and 0.27 μg/mL were examined, the optimum coating concentration of MMEC and dilution of AG-HRP were determined by checkerboard titration as described above. Then, the optimum serum dilution was determined with pooled positive bovine antiserum (120 positive bovine sera equally pooled) and pooled negative bovine serum (242 negative bovine sera equally pooled) at 1:10, 1:25, 1:50, 1:100, 1:200, 1:400, 1:800, and 1:1600 dilutions. The ratio of $OD_{450nm}$ values for the positive (P) serum to the negative (N) serum sample (P/N) was calculated. When the P/N was the highest and/or the OD value of the positive serum was slightly over 1, the reaction conditions including MMEC coating concentration, AG-HRP dilution, and serum dilution were determined to be optimal [30].

## 2.6 Determination of cut-off value, diagnostic sensitivity and specificity of MMEC/AG-iELISA for each species

The positive and negative sera of each species described above were diluted with blocking buffer at two or three titers around the optimal serum dilution determined in the above checkerboard titration. For each test, the sheep positive control (PC) serum sample and negative control (NC) serum sample were added in duplicates. Only when the average $OD_{NC}$ was smaller or equal to 0.09, while the average $OD_{PC}$ was larger or equal to 1.26 which were determined by the checkerboard titration, the test is valid. Then the S/P value for each sample was calculated as follows: S/P = $(OD_{sample} - OD_{NC})/(OD_{PC} - OD_{NC})$. The cut-off value of human, bovine, sheep, and deer sera was determined by receiver operator characteristic (ROC) analysis for maximum diagnostic sensitivity and specificity. In addition, the cut-off value of monkey sera was alternatively defined as the mean S/P value of the negative sera plus 3 standard deviations (SD) with 99% confidence due to the insufficient positive sera [24].

## 2.7 Evaluation of analytical sensitivity and specificity of MMEC/AG-iELISA

One two-fold serially diluted positive serum of cattle with *M. bovis* infection was tested in parallel with both MMEC/AG-iELISA and the IDEXX kit. Similarly, two-fold serially diluted positive serum of sheep, sika deer, roe deer, and macaque monkey with *M. bovis* or *M. caprae* infection were tested in parallel with MMEC/AG-iELISA and the INGEZIM kit. In addition, cattle antisera against *M. paratuberculosis*, *B. abortus*, Bovine herpes virus 1, and BVDV were used to evaluate the analytical specificity of MMEC/AG-iELISA.

## 2.8 Application of MMEC/AG-iELISA for the serological diagnosis of TB

To compare MMEC/AG-iELISA and commercial kits, all the bovine positive and negative sera were tested in parallel with MMEC/AG-iELISA and the IDEXX kit; all the sheep positive and negative sera were tested in parallel with MMEC/AG-iELISA and the INGEZIM kit. In addition, 145 sheep or goat sera collected in Gansu province were tested with MMEC/AG-iELISA for evaluation of the seroprevalence of sheep or goat TB in Gansu province between 2019 and 2020, 60 of them were collected in the summer of 2019, and 60 of them were collected in the summer of 2020, while 25 of them were collected in the winter of 2020.

### 2.9 Statistical analyses

By testing the antibody positive and negative sera, the ROC curve was constructed for each species. Then the area under the curve (AUC) (Area ± SE), cut-off values, diagnostic sensitivity, and diagnostic specificity of MMEC/AG-iELISA with 95% CI were determined by the software built in Graphpad 8.3 (GraphPad Software, Inc, USA). Kappa value was determined by SPSS 25 (International Business Machines Corporation, America) as follows: poor (<0.20), fair (0.20–0.40), moderate (0.41–0.60), good (0.61–0.80) and very good (0.81–1.00) [31]. Chi-square test was used to determine the difference between the positive and negative proportions with SPSS 25. The difference is significant when $p < 0.05$, and is extremely significant when $p < 0.01$.

## 3 Results

### 3.1 Development of MMEC/AG-iELISA

The recombinant fusion protein MMEC was expressed in *E. coli* and purified to a stocking concentration of 0.57 mg/mL, and confirmed to be a single band with a molecular mass of about 75 kDa in the gel after 10% SDS-PAGE (Fig 1A). Meanwhile, the titer of AG-HRP was determined as $1 \times 2^{21}$ with a stocking concentration of 4.02 mg/mL (Fig 1B). Through the checkerboard titration, the optimum coating concentration of MMEC was 50 ng/mL, the final dilution of AG-HRP was 0.4 μg/ mL, and serum dilution of 1:100 (S2 and S3 Tables). In the practical test for the serum samples from different species, the dilutions of serum samples were finely adjusted around 1:100 and the one was selected with the highest sensitivity and specificity. In detail, 120 positive bovine sera and 242 negative bovine sera were diluted at 1:50 and 1:100; 46 positive sheep sera and 113 negative sheep sera were diluted at 1:25, 1:50, and 1:100; 54 positive deer sera and 146 negative deer sera were diluted at 1:25, 1:50, and 1:100; 2 positive

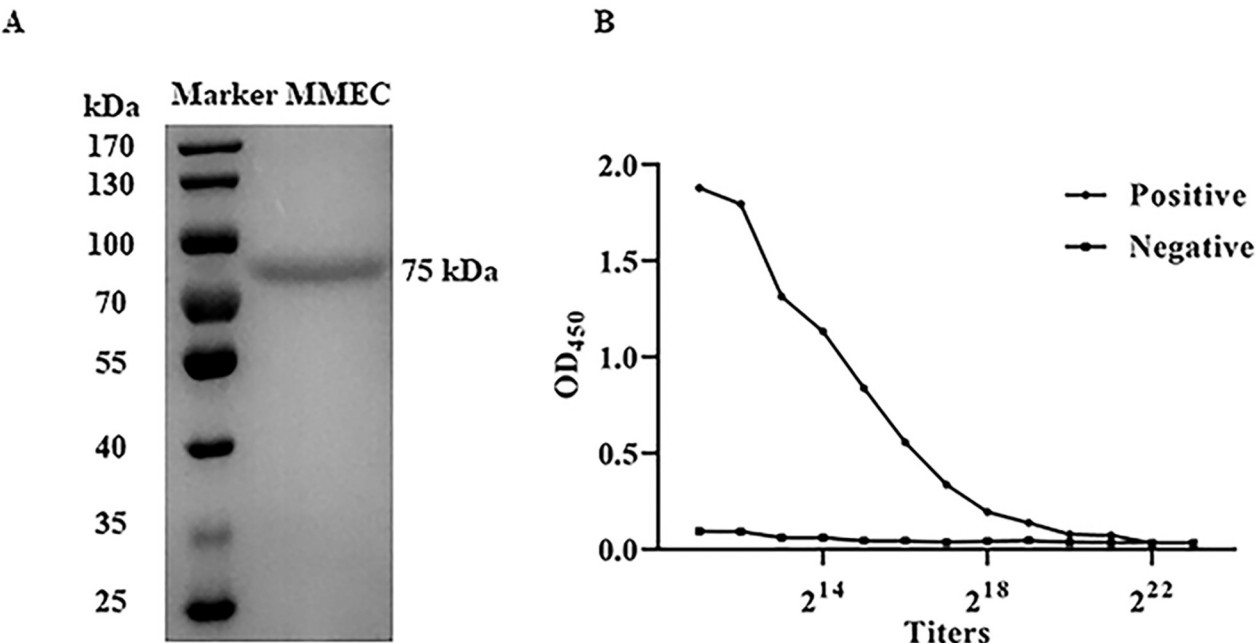

**Fig 1. Preparation of the coating fusion antigen and HRP-labeled fusion protein AG.** A: The coated fusion antigen MMEC made from MPB70, MPB83, ESAT6, and CFP10 with the size of 75 kDa was checked by 10% SDS-PAGE; B: The titers of HRP-labeled fusion protein AG made from *Staphylococcal* protein A and *Streptococcal* protein G was determined by iELISA.

monkey sera and 84 negative monkey sera were diluted at 1:25, 1:50, and 1:100; 24 positive human sera and 233 negative human sera were diluted at 1:100, 1:200, and 1:500. Then all the pre-diluted sera were tested with MMEC/AG-iELISA to further determine the parameters of this assay.

### 3.2 Detection of bTB with MMEC/AG-iELISA

By using 120 bovine antibody positive sera and 242 negative sera which were confirmed by the comparative intradermal tuberculin test, (*in vitro*) blood IFN-γ release test, and ELISA kits, this MMEC/AG-iELISA for serological detection of bTB was optimized to be as follows: serum dilution 1:50, the cut-off value of S/P 0.30 at the maximum AUC of 1 ± 0.00 (95% CI: 1.00, 1.00) ($p$<0.01) (Fig 2A) (S4 Table). Then the diagnostic sensitivity and specificity were 100.00% (95% CI: 96.90%, 100.00%) and 100.00% (95% CI: 98.44%, 100.00%), respectively. When the sera were diluted at 1:100, the cut-off value of S/P 0.24 at the maximum AUC of 0.995 ± 0.003 (95% CI: 0.99, 1.00) ($p$<0.01), then the diagnostic sensitivity and specificity were 99.59% (95% CI: 97.70%, 99.98%) and 95.00% (95% CI: 89.52%, 97.69%), respectively, (Fig 2B) (S4 Table).

Then the cattle antisera to pathogens such as *M. paratuberculosis*, Bovine herpesvirus 1, *Brucella abortus*, and BVDV were tested by this MMEC/AG-iELISA and determined to be negative based on their S/P values of 0.114, 0.011, -0.010, and 0.089, respectively, which were less than the cut-off value of 0.30.

To compare its analytical sensitivity with the commercial IDEXX kit, one cattle positive serum was two-fold serially diluted and tested with both tests. The results indicated that the highest serum dilution for MMEC/AG-iELISA was $1.6 \times 2^{14}$, which was the same as that for the IDEXX kit test (S5 Table).

Further, all the 120 positive and 242 negative bovine sera to bTB were tested in parallel with MMEC/AG-iELISA and the IDEXX kit. The results indicated the observed agreement is

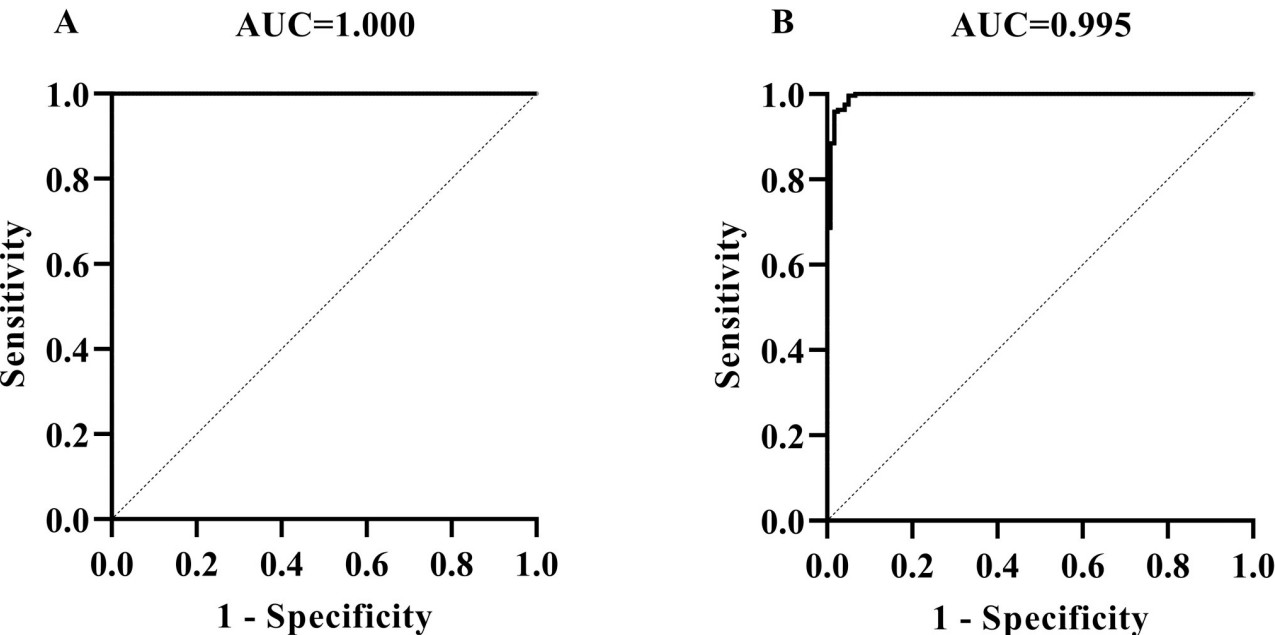

**Fig 2. Determination of cut-off values, diagnostic sensitivities, and specificities of MMEC/AG-iELISA for *M. bovis* caused bTB by using the maximum AUC.** The ROC curves of MMEC/AG-iELISA for detecting antibodies in serum samples at dilutions of 1:50 (A) and 1:100 (B).

100.00% (362/362) and Kappa value is 1, suggesting a very good agreement between these two methods (S6 Table).

Finally, 17 sera without antibody background but confirmed as *M. bovis* infection were detected by both tests. These 17 sera were collected from the cattle which had gross pathological change and confirmed as *M. bovis*-infection by mycobacterial culture and PCR typing (S1 Table), of which, 12 were determined as positive by MMEC/AG-iELISA with a sensitivity of 70.59% (95% CI: 44.04%, 89.69%), while only 9 of them were determined as positive by the IDEXX kit with a sensitivity of 52.94% (95% CI: 27.81%, 77.02%) (S1 Table). Although there is no significant difference between them ($p = 0.481$), the findings indicate this MMEC/AG-iELISA has a tendency of high sensitivity for serological diagnosis of bTB.

### 3.3 Detection of sheep TB with MMEC/AG-iELISA

By using 113 negative sheep sera from intradermal comparative test negative sheep and 46 positive sheep sera from sheep immunized with PPD-B and determined as positive by PPD-E-LISA, the sheep TB diagnosis was optimized to be as follows: serum dilution 1:50, S/P cut-off value 0.22 at the maximum AUC of $1 \pm 0.00$ (95% CI: 1.00, 1.00) ($p<0.01$) (Fig 3B) (S7 Table). Under this condition, all the positive and negative sera were correctly detected by MMEC/AG-iELISA and therefore the diagnostic sensitivity and specificity were 100.00% (95% CI: 95.00%, 100.00%) and 100.00% (95% CI: 96.80%, 100.00%), respectively. When the serum samples were diluted to 1:25, the cut-off value of S/P 0.41 at the maximum AUC of $0.999 \pm 0.001$ (95% CI: 0.997, 1.000) ($p<0.01$), then the diagnostic sensitivity and specificity were 100.00% (95% CI: 96.71%, 100.00%) and 97.83% (95% CI: 88.66%, 99.89%), respectively (Fig 3A) (S7 Table). When the serum samples were diluted to 1:100, the cut-off value of S/P 0.024 at the maximum AUC of $0.97 \pm 0.01$ (95% CI: 0.94, 0.99) ($p<0.01$), then the diagnostic sensitivity and specificity were 82.30% (95% CI: 74.24%, 88.24%) and 97.83% (95% CI: 88.66%, 99.89%), respectively (Fig 3C) (S7 Table).

Meanwhile, all the positive and negative sheep sera were tested with a commercial double antigen sandwiched INGEZIM kit. As a result, although all the negative sera were correctly detected by the INGEZIM kit, only 14 out of 46 sheep positive sera were successfully detected with a sensitivity of 30.43% (95% CI: 17.70%, 45.80%) (S8 Table), indicating that the INGEZIM kit test was significantly less sensitive than MMEC/AG-iELISA for test of *M. bovis* PPD induced antibodies in sheep ($p < 0.001$).

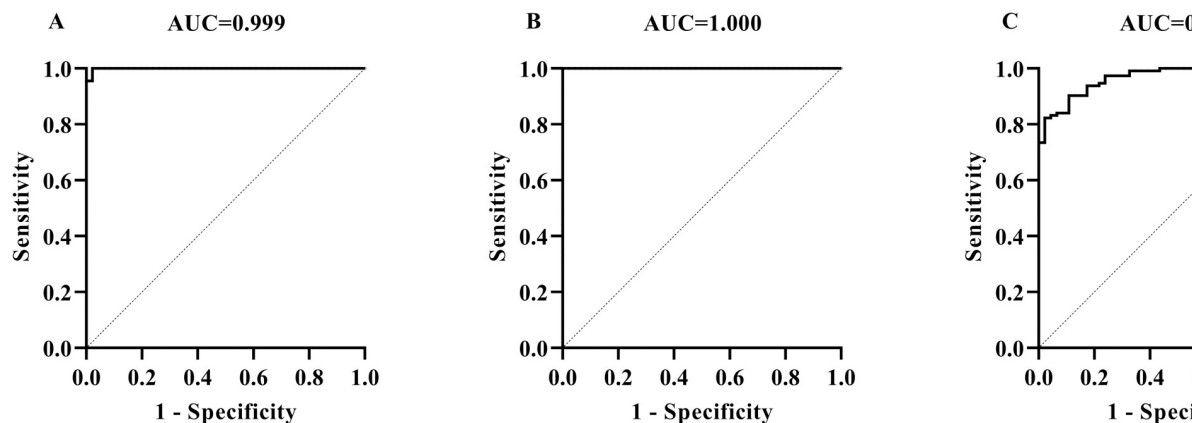

**Fig 3. Determination of cut-off values, diagnostic sensitivities, and specificities of MMEC/AG-iELISA for sheep TB caused by *M. bovis* infection by using the maximum AUC.** The ROC curves of MMEC/AG-iELISA for detecting antibodies in serum samples at dilutions of 1:25 (A), 1:50 (B), and 1:100 (C).

**Table 2. Seroprevalence of sheep and goats TB for the samples from Gansu province.**

| Sampling seasons | No. positive/ No. tested | Seroprevalence (95% CI: down, upper) |
| --- | --- | --- |
| Summer in 2019 | 9/60 | 15.00% (7.10%, 26.57%) |
| Summer in 2020 | 8/60 | 13.33% (5.94%, 24.59%) |
| Subtotal for Summer | 17/120 | 14.17% (8.47%, 21.71%) |
| Winter in 2020 | 0/25 | 0.00% (0.00%, 13.72%) |
| Total | 17/145 | 11.72% (6.98%, 18.11%) |

Then one positive sheep serum was two-fold serially diluted and tested with MMEC/AG-iELISA and the INGEZIM kit. The results indicated that with MMEC/AG-iELISA, the highest serum dilution was 1:3200, which was 16 folds higher than that with commercial INGEZIM kit (1:200) (S9 Table).

In addition, 145 sheep and goat sera clinically collected from Gansu province between 2019 and 2020 were tested with MMEC/AG-iELISA, the serological prevalence of sheep and goats TB in Gansu province was 11.72% (95% CI: 6.98%, 18.11%). Further the prevalence was 15.00% (95% CI: 7.10%, 26.57%) in the summer of 2019, 13.33% (95% CI: 5.94%, 24.59%) in the summer of 2020, higher than that in the winter of 2020 which was 0.00% (95% CI: 0.00%, 13.72%), although there was no significant difference in the seroprevalence between the summer and winter ($p = 0.08$) (Table 2).

To determine the capacity of MMEC/AG-iELISA to differentiate MTBC and nontuberculous mycobacteria (NTM) infection in sheep, the comparative tests were performed by using the intradermal comparative test, MMEC/AG-iELISA and PPD-ELISA for 26 sheep that were defined as NTM infection by the intradermal comparative test because the increase of skin fold thickness reactive to avian PPD was larger than that to bovine PPD, and larger than 4 mm. As a result, both MMEC/AG-iELISA and PPD-ELISA determined these animals were negative to TB (S10 Table).

## 3.4 Detection of cervid TB with MMEC/AG-iELISA

The samples used in this part included 52 positive sika deer sera and 146 negative sika deer sera determined by PPD-ELISA and the INGEZIM kit; 1 positive roe deer serum to *M. caprae* infection determined by clinical symptoms, mycobacterial culture, and PCR typing (Fig 4); 1 Pere David's deer positive serum to *M. bovis* infection determined by clinical symptoms, mycobacterial culture, and PCR typing (Fig 4). The cervid TB diagnosis was optimized to be as

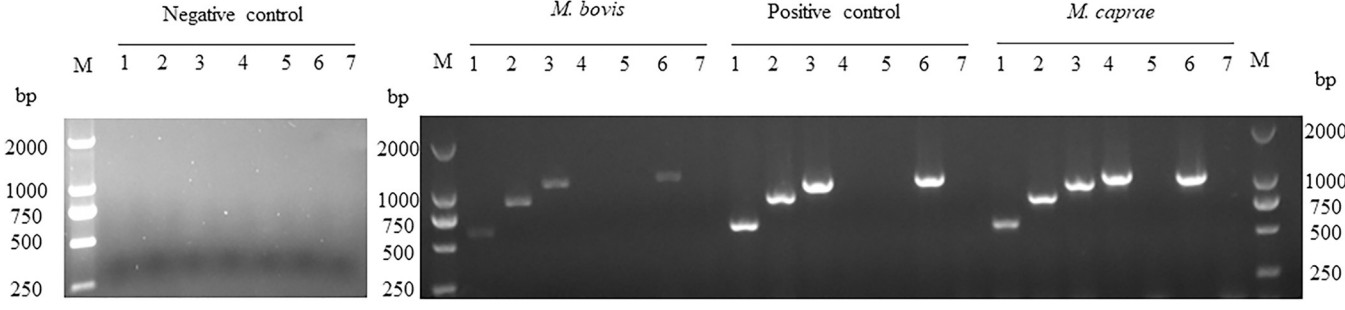

**Fig 4. Multiplex PCR typing of the mycobacterium isolates from deer species.** Negative control was deionized water; *M. bovis* was isolated from the tuberculous nodule in the lung of a Pere David's deer; Positive control was *M. bovis* ATCC19210; *M. caprae* was isolated from the abscesses in the right submaxillary face of a roe deer. Lanes: 1, *16S Rrna* (543 bp); 2, Rv0577 (786 bp); 3, IS*1561'* (943 bp); 4, Rv1510 (1033 bp); 5, Rv1970 (1116 bp); 6, Rv3877/8 (999 bp); 7, Rv3120 (404 bp); M, 2000 bp DNA ladders.

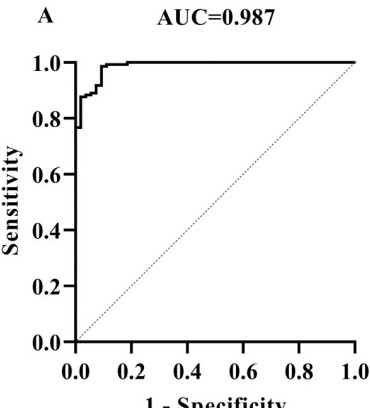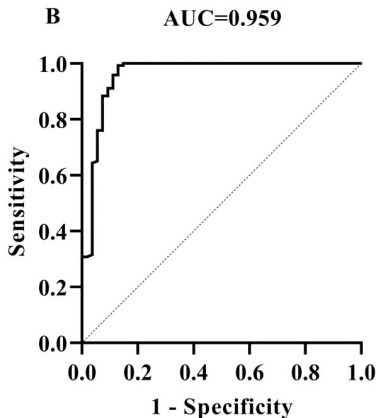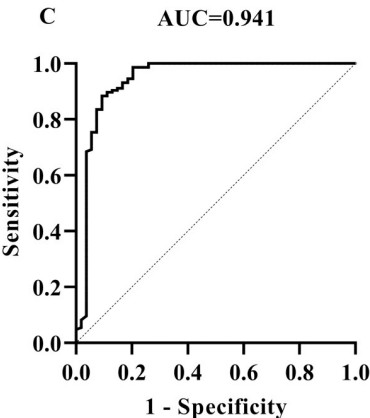

**Fig 5. Determination of cut-off values, diagnostic sensitivities, and specificities of MMEC/AG-iELISA for cervid TB caused by *M. bovis* or *M. caprae* infection by using the maximum AUC.** The ROC curves of MMEC/AG-iELISA for detecting antibodies in serum samples at dilutions of 1:25 (A), 1:50 (B), and 1:100 (C).

follows: serum dilution 1:25, S/P cut-off value 0.54 at the maximum AUC of 0.99 ± 0.01 (95% CI: 0.97, 1.00) ($p$<0.01), and diagnostic sensitivity and specificity of 90.74% (95% CI: 80.09%, 95.98%) and 98.63% (95% CI: 95.14%, 99.76%), respectively (Fig 5A) (S11 Table). When the serum samples were diluted to 1:50, the cut-off value of S/P 0.45 at the maximum AUC of 0.96 ± 0.02 (95% CI: 0.92, 1.00) ($p$<0.01), then the diagnostic sensitivity and specificity were 99.32% (95% CI: 96.22%, 99.96%) and 87.04% (95% CI: 75.58%, 93.58%), respectively (Fig 5B) (S11 Table). When the serum samples were diluted to 1:100, the cut-off value of S/P 0.06 at the maximum AUC of 0.94 ± 0.03 (95% CI: 0.89, 0.99) ($p$<0.01), then the diagnostic sensitivity and specificity were 88.36% (95% CI: 82.14%, 92.60%) and 90.74% (95% CI: 80.09%, 95.98%), respectively (Fig 5C) (S11 Table).

Then one positive sika deer serum infected with *M. bovis* was two-fold serially diluted and tested with MMEC/AG-iELISA and the commercial kit. The results indicated that with MMEC/AG-iELISA, the highest serum dilution was 1:1600, which was the same as that with the INGEZIM kit test (S12 Table).

Further, both assays were used to test one roe deer serum infected with *M. caprae*. The results indicated that with MMEC/AG-iELISA, the titer of this serum was 1:50. However, this serum was determined to be negative by the INGEZIM kit (S13 Table). For the other three serum samples collected from other three roe deer cohabited with the *M. caprae* infected roe deer, all three sera were determined to be positive by MMEC/AG-iELISA, but only two of them were determined to be positive by the INGEZIM kit.

## 3.5 Detection of TB in monkeys with MMEC/AG-iELISA

For monkey serum test, 84 negative macaque monkey sera and two positive sera determined by the upper eyelid PPD intradermal test, PPD-ELISA, and the INGEZIM kit were used at 1:25, 1:50, and 1:100 dilution. The cut-off S/P values were determined by the mean + 3 SD of negative samples to be 0.21, 0.17 and 0.11 respectively at three dilutions, then the corresponding diagnostic specificities were 97.62% (95% CI: 91.66%, 99.71%), 98.81% (95% CI: 93.54%, 99.97%), and 97.62% (95% CI: 91.66%, 99.71%) (S14 Table). Therefore, the serum dilution 1:50 and cut-off S/P value 0.17 were selected. Then, the two positive sera were determined as positive at these conditions, and the diagnostic sensitivity was 100.00% (95% CI: 15.81%, 100.00%).

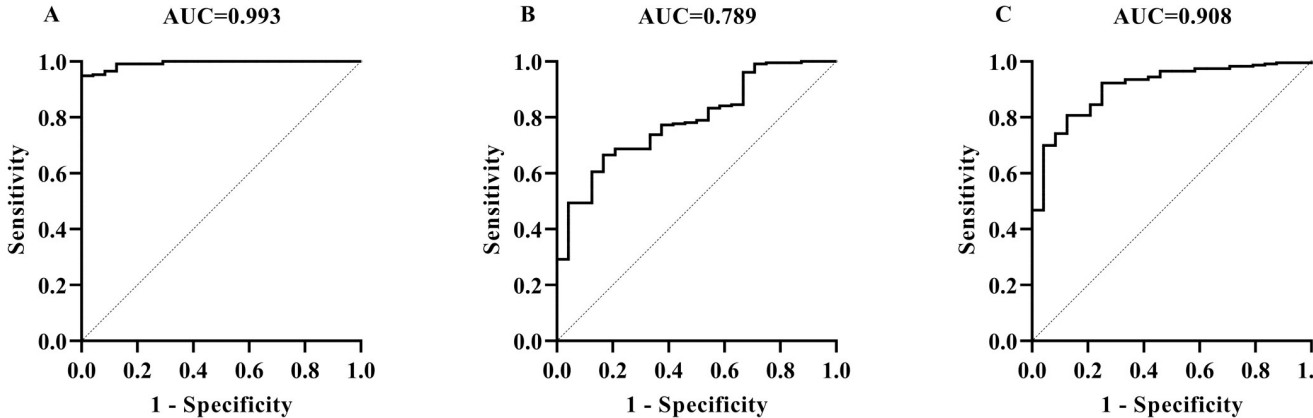

**Fig 6. Determination of cut-off values, diagnostic sensitivities, and specificities of MMEC/AG-iELISA for hTB caused by *M. tb* infection by using the maximum AUC.** The ROC curves of MMEC/AG-iELISA for detecting antibodies in serum samples at dilutions of 1:100 (A), 1:200 (B), and 1:500 (C).

Then one positive macaque monkey serum sample was two-fold serially diluted and tested in parallel with MMEC/AG-iELISA and the commercial INGEZIM kit. The results indicated that with MMEC/AG-iELISA, the highest serum dilution was 1:400, however, this serum was determined to be negative by the INGEZIM kit (S15 Table).

## 3.6 Detection of hTB with MMEC/AG-iELISA

By using 24 positive sera collected from the TB patients previously diagnosed as *M. tb* infection and 233 negative sera collected from healthy volunteers, the parameters for hTB diagnosis were determined. When serum dilutions were 1:100, 1:200 and 1:500, the cut-off values of S/P were 0.19, 0.08, and 0.03 at the maximum AUCs of 0.99 ± 0.00 (95% CI: 0.99, 1.00) ($p<0.01$), 0.79 ± 0.04 (95% CI: 0.71, 0.87) ($p<0.01$), and 0.91 ± 0.03 (95% CI: 0.85, 0.96) ($p<0.01$), then the diagnostic sensitivities and specificities were 100.00% (95% CI: 86.82%, 100.00%) and 94.85% (95% CI: 91.22%, 97.03%), 66.52% (95% CI: 60.24%, 72.27%) and 83.33% (95% CI: 64.15%, 93.32%), and 80.69% (95% CI: 75.14%, 85.24%) and 87.50% (95% CI: 69.00%, 95.66%), respectively. Therefore, the cut-off value of S/P 0.19 and serum dilution 1:100 were selected (Fig 6) (S16 Table).

## 4 Discussion

Since TB is an important zoonotic disease, the maintenance of MTBC infection in animals is a great barrier for achieving this goal of the End-TB strategy. Development of a detection technique for different live species is of significance in improving surveillance efficiency and control effect of a One Health strategy. In the current study, the universal MMEC/AG-iELISA was developed and demonstrated to have the potential in serological diagnosis of TB in humans, cattle, sheep, cervids, and monkeys.

To evaluate the performance of MMEC/AG-iELISA in serological detection of *M. bovis* infection in cattle, two sets of samples were used. One set was the serum samples with a known antibody background. For this set of samples, a complete agreement was observed between our MMEC/AG-iELISA and the commercial IDEXX kit. The very high agreement might be attributed to the strict selection of the samples because all the positive and negative samples were commonly determined by two or more tests. Further, we tested the other set of 17 serum samples from the cattle without known antibody background but identified to be infected with *M. bovis* through bacterial culture and genotyping which is considered as the gold standard method. The results showed that this MMEC/AG-iELISA had a higher sensitivity 70.59% (95%

CI: 44.04%, 89.69%) than its commercial counterpart 52.94% (95% CI: 27.81%, 77.02%). It might be attributed to the differential antigens used in these two methods. The current MMEC/AG-iELISA uses the fusion protein MMEC that consists of MPB70, MPB83, ESAT6, and CFP10. Among them, ESAT6 and CFP10 are well-known as the early secreted proteins during MTBC infection. On the other hand, the IDEXX kit uses only MPB70 and MPB83 as its coating protein and could miss the detection of early infection. Previously, inconsistent diagnostic sensitivities of the IDEXX kit such as 74% in Great Britain, 69% in Ireland, 46% in the United States, and 40% in New Zealand were reported regarding the samples diagnosed with mycobacterial methods. The inconsistent results were probably affected by the infection stages of the sampled animals because antibody development is usually at the late stage of MTBC infection [32]. Of course, more samples are needed to investigate the reason causing this difference in sensitivities. Unfortunately, it is usually difficult to get sufficient live cattle diagnosed with *M. bovis* infection for serum antibody detection. In addition, the potential reason for no antibodies being detected in five sera by MMEC/AG-iELISA might probably be that the infection was at the very early stage when the antibodies wouldn't have been significantly developed yet [28].

Although *M. tb* infection in animals has been increasingly reported in different countries in recent years [11,12,33], it is difficult to collect serum samples from confirmed *M. tb*-infected animals due to the difficulties in mycobacterial detection and genotyping for live animals. As an alternative, we used the positive and negative hTB serum samples and confirmed that this new method had a sensitivity as high as 100.00% (95% CI: 86.82%, 100.0%) and specificity as high as 94.85% (95% CI: 91.22%, 97.03%) for diagnosis of *M. tb* infection. Different sensitivities for the serological tests of hTB were previously reported, which might be attributed to the patients' different disease stages and severity degrees of TB [34,35]. Theoretically, MMEC/AG-iELISA could also be used for detecting animal *M. tb* infection. However, the performance of MMEC/AG-iELISA for serological diagnosis of *M. tb* infection in animals needs to be further evaluated in the future.

The tuberculosis of sheep can be caused by either *M. bovis* or *M. caprae* [13,36], but TB in sheep and goats is usually neglected likely due to its short lifetime of sheep and goats, less intensive farming system, and not as close contact or relationship to humans as dairy cows. In addition, it is difficult to collect serum samples from live sheep and goats diagnosed as *M. bovis* or *M. caprae* infection. As an alternative, we used the antibody positive serum samples developed by immunization of PPD-B and confirmed by PPD-ELISA. To evaluate the performance of MMEC/AG-iELISA, we compared the detection results of MMEC/AG-iELISA and the only available commercial ELISA kit (INGEZIM kit). Although the two methods were in complete agreement as to the detection of the negative samples, the sensitivity of our MMEC/AG-iELISA was significantly higher than that of the INGEZIM kit in the detection of positive samples ($p<0.001$). The most likely reason might be that the INGEZIM kit used only MPB83 as the capturing antigen. Furthermore, to determine whether this MMEC/AG-iELISA could be used in the detection of natural TB in sheep and goats, 145 random serum samples of sheep and goats were clinically collected from Gansu province and tested. An unexpected high seroprevalence of TB in sheep and goats (11.72%, 95% CI: 6.98%, 18.11%) was obtained between 2019 and 2020 with seasonal variation indicating the surveillance and control of TB in sheep and goats should be conducted in China. To our best knowledge, this is the first report on a serological survey on TB in sheep and goats. However, more samples from wider areas should be covered in future investigations to elucidate the epidemiological status of TB in sheep and goats in China. For the 113 sheep which were diagnosed as TB negative with the intradermal comparative test, 26 of them were considered as NTM infection as the increase in skin-fold thickness at the avian injection site was greater than 4mm, and greater than that in the bovine injection site. All the sera of the 26 sheep were tested by MMEC/AG-iELISA and PPD-ELISA,

and the results were negative (S10 Table). Therefore, NTM infection wouldn't affect the test results of MMEC/AG-iELISA.

Since TB in wildlife poses a great threat to humans and domestic animals, surveillance of wildlife TB is important. Among wildlife species, cervids and monkeys are very susceptible hosts and close to both humans and domestic animals [37–39]. However, wildlife can be very aggressive and are difficult to capture and handle. Therefore, conventional methods such as skin test including catching animals, measuring skin-fold thickness, injecting PPD, and measuring the increase of skin-fold thickness 72 h later is usually not practical. Therefore, developing an *in vitro* blood/serum test with high accuracy is critical for the surveillance of wildlife TB. This current study confirmed this test has high sensitivity and specificity for sika deer in serological diagnosis of *M. bovis* infection which is in line with the other serology tests [40,41]. Similarly, this method was demonstrated to work well for TB testing in macaque monkeys. However, because only two positive samples were collected during the period of this study, the cut-off value for TB serological tests in the monkeys was determined as the mean S/P value of the negative sera plus 3 SD [24]. To our knowledge, wild monkeys might have a high prevalence of TB. Since we checked the monkeys raised as the experimental animals and these monkeys have experienced several rounds of quarantine and testing, the TB prevalence in these monkeys is low. Therefore, more samples, especially positive samples, should be collected and tested to further evaluate the performance of this technique in the future.

Although *M. caprae*-caused TB has been added to the OIE disease list, *M. caprae* infection is really neglected and greatly underestimated. In addition to sheep and goats, which are susceptible hosts of *M. caprae* [42], cervids such as red deer are natural hosts and maintenance reservoirs of *M. caprae* [43]. Fortunately, we have identified one case of *M. caprae* infection in roe deer by clinical symptoms, mycobacterial isolation, and genotyping through PCR and genome sequencing, and confirmed MMEC/AG-iELISA could be used for serological diagnosis of deer *M. caprae* infection.

## 5 Conclusions

In conclusion, we established a universal MMEC/AG-iELISA for serological diagnosis of MTBC infection in multiple hosts including cattle, sheep/goats, cervids, and humans mainly caused by *M. bovis* and *M. tb* with high sensitivity and specificity. Therefore, it would be a promising application in serological TB surveillance for multiple animal species thereby providing evidence for taking further action in TB control of a One health strategy.

## Supporting information

**S1 Table. Information on the cattle confirmed positive by mycobacterial culture and PCR and serological testing with ELISA.**
(DOCX)

**S2 Table. Determination of MMEC and AG-HRP dilutions.**
(DOCX)

**S3 Table. Determination of serum dilution.**
(DOCX)

**S4 Table. The results of MMEC/AG-iELISA in serological diagnosis of *M. bovis* infection in bovines (S/P values).**
(XLSX)

**S5 Table. Analytical sensitivity of MMEC/AG-iELISA and IDEXX kit in serological diagnosis of bovine TB caused by *Mycobacterium bovis*.**
(DOCX)

**S6 Table. Comparison of MMEC/AG-iELISA and IDEXX kit in serological detection of bTB in cattle.**
(DOCX)

**S7 Table. The results of MMEC/AG-iELISA in serological detection of sheep immunized with bovine PPD (S/P values).**
(XLSX)

**S8 Table. Comparison of MMEC/AG-iELISA and INGEZIM kit in the serological detection of sheep TB.**
(DOCX)

**S9 Table. Analytical sensitivity of MMEC/AG-iELISA and INGEZIM kit in detection of sheep TB caused by *Mycobacterium bovis*.**
(DOCX)

**S10 Table. Comparative tests with intradermal comparative test, MMEC/AG-iELISA, and PPD-ELISA to differentiate MTBC and NTM infected sheep.**
(DOCX)

**S11 Table. The results of MMEC/AG-iELISA in serological diagnosis of cervid TB (S/P values).**
(XLSX)

**S12 Table. Analytical sensitivity of MMEC/AG-iELISA and INGEZIM kit in the detection of sika deer TB caused by *Mycobacterium bovis*.**
(DOCX)

**S13 Table. Analytical sensitivity of MMEC/AG-iELISA and INGEZIM kit in the diagnosis of roe deer TB caused by *Mycobacterium caprae*.**
(DOCX)

**S14 Table. The results of MMEC/AG-iELISA in serological diagnosis of monkey TB (S/P values).**
(XLSX)

**S15 Table. Analytical sensitivity of MMEC/AG-iELISA and INGEZIM kit in the detection of macaque monkey TB caused by *Mycobacterium bovis*.**
(DOCX)

**S16 Table. The results of MMEC/AG-iELISA in serological diagnosis of *M*. *tb* caused human TB (S/P value).**
(XLSX)

**S1 Raw images.**
(PDF)

## Acknowledgments

We thank all staff members from Gansu Agricultural University, Huazhong Agricultural University, and Tuberculosis department, Wuhan Medical Treatment Center who provided technical support for this study.

## Author Contributions

**Conceptualization:** Yingyu Chen, Yu Yan, Kailun Zhang, Qiaoying Zeng, Aizhen Guo.

**Formal analysis:** Liang Sun, Qiaoying Zeng, Aizhen Guo.

**Funding acquisition:** Yingyu Chen, Qiaoying Zeng, Aizhen Guo.

**Investigation:** Liang Sun, Ping Yi, Li Yang.

**Methodology:** Liang Sun, Qiaoying Zeng, Aizhen Guo.

**Project administration:** Liang Sun, Ping Yi.

**Resources:** Yingyu Chen, Qiaoying Zeng, Aizhen Guo.

**Software:** Liang Sun.

**Supervision:** Yingyu Chen, Qiaoying Zeng, Aizhen Guo.

**Validation:** Liang Sun.

**Visualization:** Liang Sun.

**Writing – original draft:** Liang Sun.

**Writing – review & editing:** Yingyu Chen, Qiaoying Zeng, Aizhen Guo.

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
