## [Decision Letter · Decision Letter 0]

21 Jul 2021

PONE-D-21-17254

Serological Detection of Mycobacterium Tuberculosis Complex Infection in Multiple Hosts by One Universal ELISA

PLOS ONE

Dear Dr. Zeng,

Thank you for submitting your manuscript to PLOS ONE. After careful consideration, we feel that it has merit but does not fully meet PLOS ONE’s publication criteria as it currently stands. Therefore, we invite you to submit a revised version of the manuscript that addresses the points raised during the review process.

It was reviewed by two experts in the field, and they have recommended some modifications be made prior to acceptance.

I therefore invite you to make these changes and to write a response to reviewers which will expedite revision upon resubmission.

We look forward to receiving your revised manuscript.

I wish you the best of luck with your modifications.

Hope you are keeping safe and well in these difficult times.

Kind regards,

Simon Clegg, PhD

Academic Editor

PLOS ONE

Reviewers' comments:

Reviewer's Responses to Questions

**Comments to the Author**

1. Is the manuscript technically sound, and do the data support the conclusions?

Reviewer #1: Yes

Reviewer #2: Yes

2. Has the statistical analysis been performed appropriately and rigorously? 

Reviewer #1: Yes

Reviewer #2: Yes

3. Have the authors made all data underlying the findings in their manuscript fully available?

Reviewer #1: No

Reviewer #2: No

4. Is the manuscript presented in an intelligible fashion and written in standard English?

Reviewer #1: Yes

Reviewer #2: Yes

5. Review Comments to the Author

Reviewer #1: This paper describes the development of a serological assay for species of the Mycobacterium tuberculosis complex or MTB is a variety of host species. I think this work is highly relevant as accurate, easy-to-use, diagnostics for animal TB are sorely lacking but pivotal to aid the control of TB. It needs to be noted however that serology/humoral immunity in TB is not very well understood and generally serological assays are not to be used as stand-alone assays as they show highly variable test performance. This assay shows promise, but it needs to be validated in a larger set of samples to access the true test performance as there may be some selection bias in the samples used in the present study. Furthermore, for some species there were rather few samples, so this needs to be addressed as well. That being said, I think the study has relevance, was well executed and should definitely be published. Please see my comments below.

General comments:

1. I couldn't find the all the data from the study. The authors declare it is all in the figures/tables or supplemental info, but I don't see for instance the outcomes of testeing the sera at various dilutions. I think this should also be made available, to strengthen the final choice of serum dilution used in the assay.

2. Adding to that, why was it chosen to only assay 2 serum dilutions (cattle) or 3 serum dilutions (in other species) rather than a range of serum dilutions in a checkerboard titration as is standard when developing a novel assay?

3. The strength of the assay is that it is able to detect Ab to various members of MTB in a number of species. However, in some cases it may be important to know with which mycobacterium the animal is infected (perhaps this is mostly true for humans, given that patients would receive different treatment for Mtb than M bovis. Any comments on this? Also, have you considered testing serum of an animal (experimentally) infected with non-tuberculous mycobacteria (NTM) to determine the specificity of this assay further? Gcebe et al. 2016 found that ESAT6, CFP10 and MPB70 were also present in the genome/proteome of some NTM, so this could mean that animals infected with those species could test positive on this assay.

4. In the abstract, you state this assay will help controlling TB. I feel like this is too strong. Animals are not likely to be culled on the basis of this assay's outcome alone. There is much more validation and evaluation needed before this happens. So I think this language needs to be a little less strong. It is definitely a good step in the right direction. Maybe use "surveillance" or "monitoring" instead of "controlling" or say that it is a step towards that.

5. In the introduction (lines 65-67): This is not really true for animal TB. As I mentioned, humoral immunity is not that well understood and so serological assays are not used that much at all (even if a lot of research has focused on the development of these assays, mostly because the relative ease which with they can be used compared to IGRAs or the skin test. Furthermore, currently (besides humans) none of the species studied in this work are currently being vaccinated by M. bovis BCG (nor any other vaccine) because of the fact that animals vaccinated with BCG test positive on the traditional TB diagnostics.

6. In section 2.2, it is stated that the human TB patients were infected with Mtb, was this confimred using molecular techniques? Also, in the section on sheep, in the skin test, I was wondering whether the presence of clinical signs of a delayed-type hypersensitivity reaction were also noted as a positive skin test? Furthermore, it is mentioned that the sheep were subcutaneously injected twice with PPD-B to raise an antibody response. It is not mentioned where on the animal this was done? And was it done using a McLintock gun like used in skin testing or a syringe?

7. In the same section, concerning different animal species: Is it not a bit too confident to say that this assay can also be used for M. caprae on the basis of one serum from a confirmed M. caprae animal? Also, how was the causative agent in monkeys determined (in other words, was it definitely M. bovis and not Mtb?). Same for sika deer, was this confirmed M. bovis by molecular techniques?

8. In section 2.5, it is explained that the optimal concentration fo the MMEC and AG-HRP if the positive OD was close to 1 and the negative OD >10. This may be a naive question, but is this a standard approach? If not, why was this chosen as optimal? I am only aware of the method that looks at the plateauing of the OD values in checkerboard titrations.

9. In section 2.6, for the ODnc and ODpc very specific numbers are given (0.09 and 1.26, respectively) but in stead of using "=", the curly equal sign is used, which means "equals approximately". This is odd. Also if these were used as validation criteria (more appropriate term than "prerequisite"), what if the OD was not exactly that, did you disregard the results? Or did you use a range? Or was it valid as long as ODnc was smaller or equal to 0.09 and ODpc larger or equal to 1.26? If so, please use the appropriate symbols.

10. In figure 1, the product looks much larger than 75kDa?

11. In section 3.2, sera from animals that were culture-confirmed M. bovis positive are tested on the two ELISAs, but the outcomes are called prevalences. This is not the appropriate term. It is the proportion of animals testing positive in a known cohort of M. bovis positive animals. A prevalence is wen you test an entire herd/population for a disease and then report how may animals tested positive.

12. In S1 Table, animals 9, 13 and 14 are M. bovis positive on bacterial culture, but not confimred by PCR. How was bacterial culture assessed? Based on morphology, ZN staining? What growth medium was used? What PCR (gene target) was used? For 2/3 of these animals the MMEC/Ag-HRP was positive, but the IDEXX negative, for the other animals both ELISAs are positive. It is mntioned that this attests to higher Se of the MMEC assay, but could it not also be a false-positive? Or do you suspect the PCR was false-negative? Please explain.

13. In section 3.3, I'd be interested to know the difference between goats and sheep as the data is now compounded together, whereas I believe sheep may be more often infected with M. bovis and goats with M. caprae (although I could be wrong?). Also, it would be interesting to see this compared to standard diagnostics for these animals? To see if the test performance of the MMEC-AG-HRP holds up.

14. Figure 3 needs a lot more information: What samples do the different lanes in the gel represent? Where are the positive and negative controls and which lanes are they in? Which gene target(s) was/were amplified and what were the conditions for the PCR (DNA extraction method, PCR reagents used, cycling conditions etc).

More specific comments:

Abstract:

Line 23: Could be changed to '..an ante-mortem technique..." and then delete "live" from te sentence.

Line 24: Remove the words "effect based on" and change to "..employing a One Health.."

Line 27: Brackets around MMEC are missing: (MMEC). Also, I think you mean "secondary antibody" or "detection antibody". the term "second antibody" appears in several places in the manuscript. Please correct everywhere.

Line 31: Change "Besides" to "Furthermore".

Introduction:

There do not appear to be any references for the first couple of sentences? Would be worth adding some here.

Line 38: Not sure I like the term "cross-infected", consider removing "cross-" and just saying "..infected by various.."

Line 50: Change "Besides" to "Additionally".

Line 63: Change to "or are unsuitable" ("are" is missing from your sentence).

Line 70: Change "possibly used" to "possible candidate"

Line 71: Change "etc" to "among others"

Line 76: Change "..addressing it is to develop antigen.." to "..addressing this is to develop an antigen..".

Line 78: Change "furthermore" to "however" and change "..antigen, might.." to "..antigen, it might.."

Line 80: Remove "by".

Line 82: Remove "was".

Line 86: Change "calibrated" to "optimized". This occurs in several places in the manuscript and should be revised everywhere.

Section 2.2

Line 94: Change "was" to "is".

Lines 109/110: Change "shorten" (which is grammatically incorrect) to "shortened" or "abbreviated".

Line 112: 72h should be written out as it is at the start of a sentence (so "Seventy-two hours).

Section 2.4

Line 144: Change "The 100" to "A volume of 100".

Line 154: Blocking buffer is suddenly mentioned for the first time without being defined. Assume this is 1% fish gelatin in PBS?

Line 158: What is the concentration of Tween in PBS in the PBST for washing?

In this section (and maybe others) you say "the plate is added with xxx" which is a bit odd. I would suggest to change this to "xxx was added to the plate". Please change this accordingly, throughout the manuscript.

Section 2.6

"respectively" is used a lot, but it is out of place. In the first sentence talking about the bovine samples, the use of "respectively" makes it seem as though you used 1:50 for the positive samples and 1:100 for the negative samples. Whereas I assume both dilutions were used for both positive and negative samples. Then for the other species it gets even more confusing as there are now 3 dilutions. I'd just remove "respectively" from all sentences. Also, why only these dilutions as opposed to a range? And to reiterate: please also show these data.

Section 2.8

The heading "Application of MMEC/AG-iELISA to TB serological diagnosis" reads a bit funny. Maybe change it to something like "Application of MMEC/AG-iELISA for the serological diagnosis of TB".

Section 3.2

Line 223: "combinedly" is a very odd word. Revise.

Line 249: "as a result" is not used appropriately here. Revise.

Line 252: "no difference" should be "no significant difference".

Here "calibrated" is used a lot, but "optimized" is more fitting.

In table 2, in the headings of the columns, change "down" to "lower".

Section 3.4

Line 294: "Detect" should be changed to "test" ("tests" should then maybe be changed to "assays").

Section 4

Line 315: I feel like it is a bit obvious that TB impedes the End-TB strategy? Make it clear that you make TB in animals versus TB in humans and the relationship between the two.

Line 328: Change "were" to "is".

Line 342: Remove "be".

Lines 345 and 356: Change "Alternatively" to "As an alternative".

Line 348: This sentence is too long and complex. End the sentence after "M. tb infection." and then start the new sentence "Different...." (so remove "Although).

Line 364: "As a result" is not necessary here and a bit out of place.

Line 384: Change "Besides sheep and goats are susceptible.." to "Besides sheep and goats, which are susceptible.." or similar.

Reviewer #2: This was a very interesting paper which investigates the development of a multi TB species ELISA using multiple antigens on a single plate. It will be a welcomed addition to an area where simple diagnostics are sorely lacking.

Major points

I am surprised that you only test a few dilutions for the titrations. Was there a reason for this? I would have expected to see more

As a question, would the assay detect organisms such as M. avium sub. Paratuberculosis (Johnes disease)? This would be of interest too. And how does it cope with a co-infection?

I think stating that the assay will aid in disease control may be a bit over the top. Although they do slaughter based on the skin tests, it may be that this assay would supersede this, but maybe worth edging on the side of caution.

Line 23- improving the surveillance effort based on a One Health strategy (reword)

Line 25- please define MTBC in the abstract

Line 27- as the secondary antibody (reword)

Line 28- comma after then

Line 30- some values in here may be useful

Line 31- in addition may sound better than besides.

Line 38-40- could you link some of these short sentences to aid flow for the reader? Also M bovis and M caprae need to be in full on their first usage. I also feel that these may benefit from some references being inserted

Line 43- this reads a bit awkwardly, please reword

Line 49- delete the

Line 51- maybe susceptible rather than sensitive?

Line 52-53- this reads a bit awkwardly, please reword

Line 61- such as the tuberculin skin … (reword)

Line 63- either have low sensitivities… (reword)

Line 63- or are unsuitable for the …. (reword)

Line 65- this isn’t completely true for animal TB. I also don’t believe that the M. bovis vaccine is used much- but I could be incorrect.

Line 67- comma after screening

Line 67- is this M. bovis, or should it be M. tb?

Line 68- low sensitivity may sound better than lowly?

Line 70- the potential antigens may sound better?

Line 70- were these antigens used for all host species?

Line 74- limitation of the antibody test with a traditional … (reword)

Line 76- develop an antigen … (reword)

Line 79- comma after test

Line 80- delete by

Line 81- overcome the Ig species ….(reword)

Line 82- aimed to develop a single universal ….(reword)

Line 85- maybe worth saying A and G proteins of which bacteria

Line 85- bind to Ig of various host species …. (reword)

Line 94- is presented in table 1 (reword)

Line 94- how were the human positive and negative sera identified? Was this by PCR or an ELISA? Please state in here. This is also true for other species

Line 94- comma after sera

Line 98- is seropositive and TB positive the same here?

Line 99- in vitro in italics

Line 101- and were confirmed as ….(reword)

Line 103- confirmed by the comparative …(reword)

Line 103- in vitro in italics

Line 103- and with the above mentioned ELISA kits (reword)

Line 104- this is repetitive and could be removed

Line 105- B. abortus in brackets isn’t needed and can be deleted

Line 109- maybe PPD isn’t the best phrasing- perhaps just say TB? Or explain what PPD is

Line 114- positive sheep sera (reword)

Line 115- whats PPD-B?

Line 117-121- I found this unclear, so please reword

Line 122- remove were

Line 123- one positive serum sample to … (reword)

Line 128- remove both ‘were’

Table 1- could this be separated more for clarity?

Please ensure that in vitro is in italics in Table 1

Also include a dash in table one where there are blank cells

Line 136- is there a reference for that?

Line 138-139- please include manufacturers in here

Line 144- 100ul of 2omg … (reword)

Line 144- please have NaIO4 in full

Line 147- 1 mL of 2mg …(reword)

Line 148- please have NaBH4 in full

Line 150- maybe incubated or mixed rather than added

Line 154- maybe incubated or mixed rather than added

Line 155- immune with MMEC- maybe reword this as it seems unclear

Line 157- then the plate was incubated with? (maybe reword)

And 159- and 160

Line 161- is this 1M or 1 N?

Line 171- delete as

Line 171- diluted in what?

Line 182- deer sera was determined …(reword)

Line 183-184- Please reword as this reads awkwardly

Line 196- in addition may sound better than besides

Line 204 and 207- please include the manufacturer of these stats packages

Line 211- E. coli (include space)

Line 211- maybe purified to a stocking concentration?

Line 213- please reword as this doesn’t make sense

Figure 1- is that 75KDa? It looks larger than that but may be the image?

Line 223- combinedly confirmed sounds strange- please reword

Line 224- please put in vitro in brackets.

Table 2- is there any reason why cervids detection limits are lower?

Line 255- negative sheep sera (reword)

Line 256- positive sheep sera (reword)

Line 267- why only one sample used here? I would like to see more used for confirmation

Line 282- this reads awkwardly, please reword

Line 287- I feel that figure 3 needs more information0 what is it showing, whats in each lane, controls? Sizes, targets etc as well as methodology for production

Line 313- why no dilution series in here?

Line 318- of a One Health Strategy (reword)

Line 322- with a known antibody …(reword)

Line 327- infected with M. bovis (reword)

Line 334- sensitivities of the IDEXX …(reword)

Line 340- space between M and bovis

Line 342- would not have (replace weren’t be)

Line 349- will this not depend on the patients disease status?

Line 354- short lifetime- sheep or the disease?

Line 355- close contact or relationship to humans maybe?

Line 355- serum samples from live …(reword)

Line 361- of the INGEZIM kit …(reword)

Line 361- maybe the most probably reason? Or most likely reason?

Line 362- of the INGEZIM kit …(reword)

Line 366- with seasonal variation ….(reword)

Line 370- delete the

Line 372- maybe ‘wildlife can be very aggressive and are difficult to capture and handle, and this test?

Line 375- in vitro in italics

Line 378- TB testing in the …(reword)

Line 379- is monkey TB rarer than the other types?

Line 381- comma after both samples

Line 383- this doesn’t make sense- M. caprae infected TB? Please reword

Line 384- although may sound better than besides?

Line 392- space between M. bovis and M. tb

Line 393- of a One Health strategy (reword)

6. PLOS authors have the option to publish the peer review history of their article (what does this mean?). If published, this will include your full peer review and any attached files.

Reviewer #1: No

Reviewer #2: No

---

## [Author Response · Author response to Decision Letter 0]

23 Aug 2021

Dear Editor, 

Thank you very much for your kind and useful comments. We have read them carefully and responded to them point by point. Briefly, we added the detailed information for the serum collection, the pictures of the positive and negative controls to the seven genes in multiplex PCR typing (Fig.4 in the revised version), and ROC generated at different dilutions of the serum samples of each species. Furthermore, we re-ran the 10% SDS-PAGE to check the size of MMEC expected to 75kDa. These additional data increase the supplementary tables from 9 to 16 (S1-S16 Tables), and the figures from 3 to 6 (Fig.1 to Fig.6). The original uncropped and unadjusted images are in supporting information (S1_raw_images). We really appreciated the critical and kind comments from you and the reviewers and believe the revised version is greatly improved. I hope it will meet your requirement for publication. 

If you have any more questions, please feel free to contact me. 

Thanks again and wish all the best, 

Respectively yours, 

Qiaoying Zeng

---

## [Editor Report · Decision Letter 1]

14 Sep 2021

Serological Detection of Mycobacterium Tuberculosis Complex Infection in Multiple Hosts by One Universal ELISA

PONE-D-21-17254R1

Dear Dr. Zeng,

We’re pleased to inform you that your manuscript has been judged scientifically suitable for publication and will be formally accepted for publication once it meets all outstanding technical requirements.

Kind regards,

Simon Clegg, PhD

Academic Editor

PLOS ONE

Additional Editor Comments:

Many thanks for resubmitting your manuscript to PLOS One

As you have addressed all the comments and the manuscript reads well, I have recommended it for publication

You should hear from the Editorial Office shortly.

It was a pleasure working with you and I wish you the best of luck for your future research

Hope you are keeping safe and well in these difficult times

Thanks

Simon

---

## [Editor Report · Acceptance letter]

29 Sep 2021

PONE-D-21-17254R1 

Serological Detection of *Mycobacterium Tuberculosis* Complex Infection in Multiple Hosts by One Universal ELISA 

Dear Dr. Zeng:

I'm pleased to inform you that your manuscript has been deemed suitable for publication in PLOS ONE. Congratulations! Your manuscript is now with our production department. 

Kind regards, 

on behalf of

Dr. Simon Clegg 

Academic Editor

PLOS ONE